# Slit2 as a β-catenin/Ctnnb1-dependent retrograde signal for presynaptic differentiation

**Haitao Wu**[1,2†], **Arnab Barik**[2,3†], **Yisheng Lu**[2,3], **Chengyong Shen**[2,3], **Andrew Bowman**[2,3], **Lei Li**[2,3], **Anupama Sathyamurthy**[2,3], **Thiri W Lin**[2,3], **Wen-Cheng Xiong**[2,3,4], **Lin Mei**[2,3,4*]

[1]Department of Neurobiology, Institute of Basic Medical Sciences, Beijing, China; [2]Department of Neuroscience and Regenerative Medicine, Medical College of Georgia, Georgia Regents University, Augusta, United States; [3]Department of Neurology, Medical College of Georgia, Georgia Regents University, Augusta, United States; [4]Charlie Norwood VA Medical Center, Augusta, United States

**Abstract** Neuromuscular junction formation requires proper interaction between motoneurons and muscle cells. β-Catenin (Ctnnb1) in muscle is critical for motoneuron differentiation; however, little is known about the relevant retrograde signal. In this paper, we dissected which functions of muscle Ctnnb1 are critical by an in vivo transgenic approach. We show that *Ctnnb1* mutant without the transactivation domain was unable to rescue presynaptic deficits of *Ctnnb1* mutation, indicating the involvement of transcription regulation. On the other hand, the cell-adhesion function of Ctnnb1 is dispensable. We screened for proteins that may serve as a Ctnnb1-directed retrograde factor and identified Slit2. Transgenic expression of Slit2 specifically in the muscle was able to diminish presynaptic deficits by *Ctnnb1* mutation in mice. Slit2 immobilized on beads was able to induce synaptophysin puncta in axons of spinal cord explants. Together, these observations suggest that Slit2 serves as a factor utilized by muscle Ctnnb1 to direct presynaptic differentiation.

*For correspondence: lmei@gru.edu

†These authors contributed equally to this work

Competing interests: The authors declare that no competing interests exist.

## Introduction

The neuromuscular junction (NMJ) is a cholinergic synapse that rapidly conveys signals from motoneurons to muscle cells to control muscle contraction. It exhibits a high degree of subcellular specialization characteristic of chemical synapses and has served as a model to study synaptogenesis (*Sanes and Lichtman, 2001*; *Wu et al., 2010*; *Barik et al., 2014b*). Abnormal NMJ formation or function causes neurological disorders including congenital myasthenic syndrome and myasthenia gravis. NMJ formation requires reciprocal interactions between nerve terminals and muscle cells. In anterograde signaling, nerve-derived agrin directs post-synaptic differentiation by interacting with muscle transmembrane receptors LRP4 and MuSK, which in turn activate downstream signaling events leading to AChR concentration at the NMJ (*McMahan, 1990*; *DeChiara et al., 1996*; *Glass et al., 1996*; *Weatherbee et al., 2006*; *Kim et al., 2008*; *Zhang et al., 2008*). However, much less is known regarding retrograde mechanisms that control motor neuron survival and presynaptic differentiation.

Wnt signaling has been implicated in NMJ formation in *Caenorhabditis elegans*, *Drosophila* as well as vertebrates (*Korkut and Budnik, 2009*; *Zhang et al., 2009*; *Barik et al., 2014b*). β-Catenin (Ctnnb1) is a central component of the canonical Wnt signaling pathway, regulating transcription by associating with TCF/LEF transcription factors (*MacDonald et al., 2009*). It also regulates adhesion-dependent signaling by binding to cadherins and α-catenin (*Nelson and Nusse, 2004*). Intriguingly, when *Ctnnb1* is

**eLife digest** Motor nerves are like electrical wires that connect our spinal cord to the muscles in our body. These nerves communicate with muscles across a connection called the neuromuscular junction. To first form a neuromuscular junction, the motor nerves and muscles each produce molecular cues that tell each other to do their part to build a connection. Beta-catenin in the muscle is known to regulate motor nerve development. However, beta-catenin has two different roles: it helps to coordinate whether neighboring cells stick together, and it can regulate which genes are 'transcribed' to produce proteins. It was not known which of these roles is necessary for forming neuromuscular junctions.

Wu, Barik et al. now investigate this question by creating mice with mutant forms of beta-catenin in their muscles. Some mice had muscle beta-catenin that could not help cells stick together, and others had beta-catenin that could not control gene transcription. Only mutations that affected the ability of beta-catenin to control transcription caused abnormalities in the neuromuscular junction. However, these problems could be fixed by adding either normal beta-catenin or the mutant form that cannot help cells stick together.

Wu, Barik et al. then used molecular tools to explore which genes are turned on by beta-catenin. The experiments showed that beta-catenin causes muscle fibers to produce a protein called Slit2—a developmental cue that controls where neurons grow. Furthermore, the neuromuscular junction defects found in mice without beta-catenin in their muscles could be reduced by making the muscle fibers produce more Slit2. However, not all defects in beta-catenin mutant mice are rescued by Slit2. Future research is needed to identify other beta-catenin-controlled signals and to determine whether such a pathway is altered in neuromuscular disorders.

mutated in muscle fibers, mutant mice die neonatally, with profound presynaptic deficits such as mislocation of phrenic nerve primary branches, reduced synaptic vesicles, and impaired neuromuscular transmission (*Li et al., 2008*), suggesting that Ctnnb1 in muscle cells is necessary for presynaptic differentiation. In support of this notion are recent reports that expression of stable Ctnnb1 in muscle cells also impairs presynaptic differentiation in mutant mice (*Liu et al., 2012*; *Wu et al., 2012a*). These observations suggest that Ctnnb1 in muscle is critical for a retrograde pathway to direct nerve terminal development. However, the underlying mechanism remains unclear. The interaction of cadherins of pre- and post-synaptic membranes has been shown to be important for synapse formation (*Bamji et al., 2003*; *Bozdagi et al., 2004*; *Prakash et al., 2005*) and synaptic plasticity (*Murase et al., 2002*; *Schuman and Murase, 2003*; *Nuriya and Huganir, 2006*). NMDAR stimulation accumulates Ctnnb1 in spines, which in turn regulates induced endocytosis of N-cadherins (*Tai et al., 2007*). These observations raise questions whether the muscle Ctnnb1 regulates presynaptic differentiation via cell adhesion-dependent signaling and/or gene-expression.

In this paper, we determined which function of Ctnnb1 is required for NMJ formation by characterizing transgenic mice expressing wild-type or Ctnnb1 mutants that were impaired in transcriptional regulation or cell-adhesion signaling. Rescue experiments indicated a necessary role for the transcription activity of muscle Ctnnb1 in presynaptic differentiation. Our exploration of targets of Ctnnb1 as potential muscle-derived retrograde factors led to the identification of Slit2, an environmental cue that repels or collapses neuronal axons (*Brose et al., 1999*; *Kidd et al., 1999*). Slit2 belongs to a family of large ECM (extracellular matrix) glycoproteins known to be chemorepellent for olfactory, motor, hippocampal, and retinal axons (*Nguyen Ba-Charvet et al., 1999*; *Erskine et al., 2000*; *Niclou et al., 2000*; *Ringstedt et al., 2000*). However, Slit2 was also shown to stimulate the formation of axon collateral branches by dorsal root ganglia neurons (*Wang et al., 1999*) and positively regulate motor axon fasciculation (*Jaworski and Tessier-Lavigne, 2012*). Slit2 was able to induce clusters of synaptophysin in cultured neurons, suggesting a synaptogenic function. Expression of Slit2 specifically in muscle fibers was able to rescue NMJ deficits in Ctnnb1-mutant mice. These observations demonstrate that Ctnnb1 regulates presynaptic differentiation by a transcription-dependent mechanism and identify Slit2 as a novel retrograde factor in NMJ formation.

## Results

### Transcriptional activity of muscle Ctnnb1 is crucial for presynaptic differentiation and function

The N-terminal region of Ctnnb1 interacts with α-catenin, critical for cell adhesion. The key amino acid residues in Ctnnb1 for interaction with α-catenin have been mapped to Thr-120 and Val-122 (*Aberle et al., 1996a*, *1996b*). Mutation of these two residues to alanines prevents Ctnnb1 from binding to α-catenin (*Xu et al., 2000*). However, the mutation has no effect on binding to TCF/Lef1, and thus, does not alter the transcription by Ctnnb1 and TCF/Lef1 (*Xu et al., 2000*). On the other hand, the transcriptional regulation requires the transactivation domain (TAD) (*Molenaar et al., 1996*; *van de Wetering et al., 1997*; *Vleminckx et al., 1999*). To determine which function of *Ctnnb1* is necessary, we generated transgenic mice: *Ctnnb1*TV-AA (LSL-*Ctnnb1*TV-AA) to impair the cell-adhesion function and TAD-deleted *Ctnnb1* (LSL-*Ctnnb1*ΔTAD) to eliminate the transcriptional regulation, as well as wild-type LSL-*Ctnnb1*. Each transgene was placed downstream of a loxP-flanked transcriptional STOP (LSL) and contained the lacZ/neomycin fusion gene and triple repeats of the SV40 polyadenylation signal (*Zinyk et al., 1998*). Expression of the transgenes, which were tagged by an HA epitope in the C-terminus, was under the control of the CAG promoter (the CMV enhancer/chicken β-actin promoter) and depended on expression of a Cre recombinase (*Zinyk et al., 1998*) (*Figure 1—figure supplement 1A,B*). In HEK293 cells, co-expression of GFP-Cre, which floxed out the stop element, led to expression of *Ctnnb1*, *Ctnnb1*TV-AA, and *Ctnnb1*ΔTAD that could be detected by both anti-HA and anti-Ctnnb1 antibodies (*Figure 1—figure supplement 1C*).

To determine whether Ctnnb1 and mutants were expressed in skeletal muscles, the transgenic mice were crossed with ACTA1-Cre mice that express the Cre recombinase specifically in skeletal muscles under the control of the promoter of the human skeletal alpha-actin (HSA or ACTA1) gene (*Miniou et al., 1999*; *Li et al., 2008*; *Wu et al., 2012a*, *2012b*). As shown in *Figure 1—figure supplement 1C,D*, HA-tagged *Ctnnb1* and mutants were expressed in muscles of ACTA1-Cre::LSL-*Ctnnb1*, but not in muscles of ACTA1-Cre or mice that carried only the transgenes. Moreover, the transgene expression was muscle-specific and not detectable in other tissues including spinal cord (*Figure 1—figure supplement 1D,E*) (data not shown). ACTA1-Cre::LSL-*Ctnnb1*, ACTA1-Cre::LSL-*Ctnnb1*TV-AA, and ACTA1-Cre::LSL-*Ctnnb1*ΔTAD mice were viable and fertile and exhibited normal NMJ morphology, compared to control mice (ACTA1-Cre or *Ctnnb1*$^{fl/fl}$) (*Figure 1—figure supplement 1F*) (data not shown). No difference was observed in location of primary branches of the phrenic nerves, secondary branches, and location and size of AChR clusters at postnatal day 0 (P0) between controls and mutants (*Figure 1—figure supplement 1F*) (data not shown).

We bred the LSL-transgenic mice with ACTA1-Cre::*Ctnnb1*$^{fl/+}$ mice (ACTA1-*Ctnnb1*$^{fl/+}$) to determine whether and which transgene was able to rescue NMJ deficits by Ctnnb1 mutation. Notice that both the expression of wildtype or mutant Ctnnb1 and the deletion of endogenous *Ctnnb1* depend on the ACTA1 promoter. In control mice (*Ctnnb1*$^{fl/fl}$), primary branches of the phrenic nerve were located in the middle of muscle fibers. The secondary branches were numerous and short, many of which end with AChR clusters that were confined to the central region of muscle fibers (*Lin et al., 2000*; *Li et al., 2008*) (*Figure 1A*). However, in ACTA1-*Ctnnb1*$^{-/-}$ (i.e., CATA1::*Cre*;*Catnnb1*$^{f/f}$) mutants, the primary branches were mislocated to the tendon region close to the central cavity, the secondary branches were fewer and longer, and the region where AChR cluster distributed became wider (*Figure 1*). As expected, expression of wild-type Ctnnb1 was able to rescue these phenotypes (*Figure 1*). Interestingly, these phenotypes were also completely rescued by *Ctnnb1*TV-AA since no difference was detectable in primary branch location, secondary branch number and length, and endplate band width between ACTA1-*Ctnnb1*$^{-/-}$::LSL-*Ctnnb1*ΔTAD mice and control or ACTA1-*Ctnnb1*$^{-/-}$::LSL-*Ctnnb1* mice, indicating that the cell-adhesion function of Ctnnb1 is not necessary for presynaptic differentiation. In contrast, however, the TAD deletion mutant was unable to rescue these phenotypes, suggesting a requirement of the transcription function (*Figure 1*).

We have shown that muscle-specific ablation of Ctnnb1 disrupts presynaptic structure and function (*Li et al., 2008*; *Wu et al., 2012a*). To determine whether the presynaptic deficits could be rescued, we performed electron microscopic analysis. In control mice, axon terminals were filled with synaptic vesicles, some of which were docked on electron-dense active zones (*Figure 2*). The vesicle density in control terminals was $3.6 \pm 0.30$ vesicles/0.04 μm$^2$. In contrast, the density was reduced to $1.40 \pm 0.37$

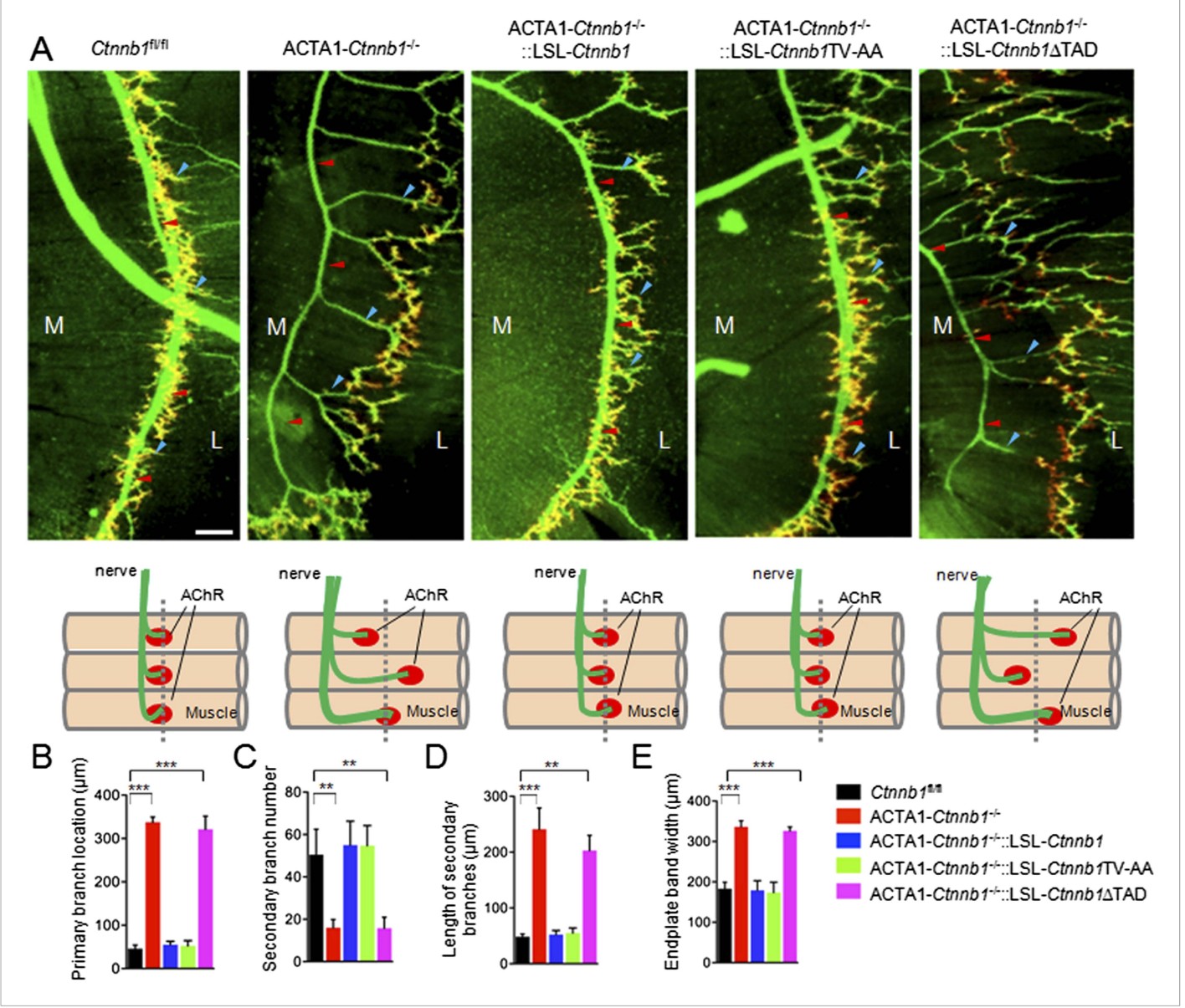

**Figure 1**. Requirement of the TAD domain to rescue presynaptic deficits in ACTA1-*Ctnnb1*$^{-/-}$ deficits. (**A**) Diaphragms of P0 mice of indicated genotypes were stained whole mount with rhodamine-conjugated α-BTX (R-BTX) to label AChR clusters and anti-NF/synaptophysin antibodies (green) to label axons and nerve terminals. Shown were left ventral areas. Red arrowhead, primary branches; blue arrowheads, secondary branches. M, medial; L, lateral. Diagrams summarizing morphological deficits. (**B–E**) Quantitative analysis of data in **A**. Data were shown as mean ± SEM; **, p < 0.01; ***, p < 0.001; One-way ANOVA; n = 4; bar, 50 μm.

The following figure supplement is available for figure 1:

**Figure supplement 1**. Generation and identification of LSL-*Ctnnb1*, LSL-*Ctnnb1*TV-AA, LSL-*Ctnnb1*ΔTAD transgenic mice.

vesicles/0.04 μm$^2$ in ACTA1-*Ctnnb1*$^{-/-}$ terminals (p < 0.001), in agreement with previous report (*Liu et al., 2012*; *Wu et al., 2012a*). As shown in *Figure 2*, the reduction in vesicle density was diminished by expressing *Ctnnb1* or *Ctnnb1*TV-AA, with 3.6 ± 0.33 and 3.99 ± 0.33 vesicle/0.04 mm$^2$ in ACTA1-*Ctnnb1*$^{-/-}$::LSL-*Ctnnb1* mice and ACTA1-*Ctnnb1*$^{-/-}$::LSL-*Ctnnb1*TV-AA mice, respectively (p < 0.001 in comparison with ACTA1-*Ctnnb1*$^{-/-}$, n = 17) (*Figure 2A,B*). These results indicated that presynaptic deficits could be rescued by wild-type *Ctnnb1* or *Ctnnb1*TV-AA, suggesting a dispensable role of the cell-adhesion function of Ctnnb1. In contrast, synaptic vesicle

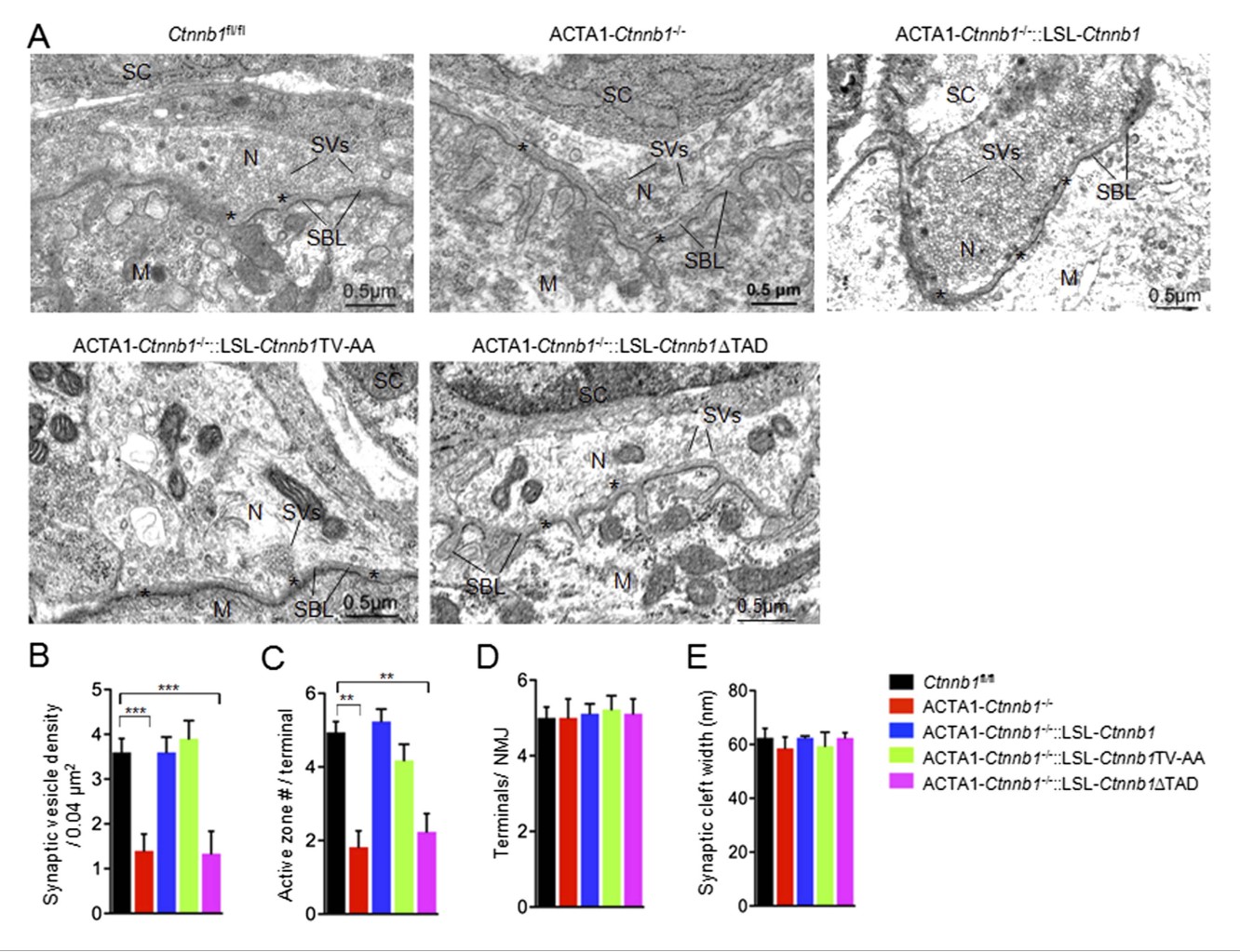

**Figure 2**. The TAD is necessary for muscle Ctnnb1 to regulate presynaptic development. Diaphragms of P0 mice were subjected to electron microscopic analysis. (**A**) Representative electron micrographic images of indicated genotypes. Asterisks, active zones; SBL, synaptic basal lamina; N, nerve terminals; M, muscle fibers; SC, Schwann cells; SV, synaptic vesicle. (**B–E**) Quantitative analysis of data in **A**. Data were shown as mean ± SEM; **, p < 0.01; ***, p < 0.001, One-way ANOVA, n = 10.

The following figure supplement is available for figure 2:

**Figure supplement 1**. Synaptic nuclear localization of Ctnnb1 and *Axin2*-nlacZ in the NMJ region.

density in ACTA1-*Ctnnb1*$^{-/-}$::LSL-*Ctnnb1*ΔTAD remained low, at 1.33 ± 0.50 vesicles/0.04 μm$^2$, which was not different from ACTA1-*Ctnnb1*$^{-/-}$ mice (p > 0.05 in comparison with control; n = 17) (*Figure 2A,B*), demonstrating inability of the ΔTAD mutant to rescue and suggesting an important role of the transcription regulation in the retrograde pathway. Similarly, active zones were fewer in ACTA1-*Ctnnb1*$^{-/-}$ mice and ACTA1-*Ctnnb1*$^{-/-}$::LSL-*Ctnnb1*ΔTAD mice, compared with control mice, ACTA1-*Ctnnb1*$^{-/-}$::LSL-*Ctnnb1* mice, and ACTA1-*Ctnnb1*$^{-/-}$::LSL-*Ctnnb1*TV-AA mice (*Figure 2A,C*). As reported previously (*Wu et al., 2012a*), *Ctnnb1* mutation in muscles did not alter the number of terminals per NMJ and the width of synaptic clefts. Expression of *Ctnnb1* wild type or mutants did not change these parameters (*Figure 2A,D,E*). These results are in agreement with the notion that muscle Ctnnb1 mainly regulates presynaptic differentiation.

In addition to morphological studies, we measured miniature end-plate potentials (mEPPs), events that are generated by spontaneous vesicle release. There was about 50% reduction in mEPP frequency in ACTA1-*Ctnnb1*$^{-/-}$ mice, compared with control mice (*Li et al., 2008*) (*Figure 3A,B*)

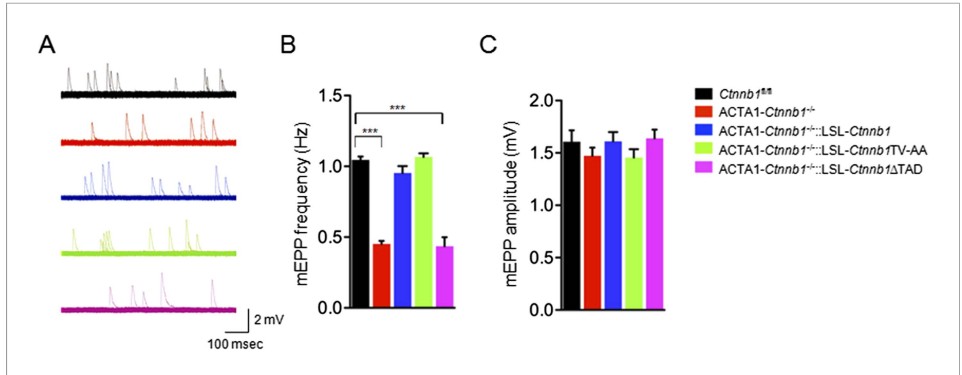

**Figure 3**. Rescue of neurotransmission deficits depends on Ctnnb1 TAD domain. miniature end-plate potentials (mEPPs) were recorded in P0 diaphragms of indicated genotypes. (**A**) Representative mEPP traces. (**B–C**) Quantitative analysis of mEPP amplitude (**B**) and frequency (**C**). Data were shown as mean ± SEM; ***, p < 0.001; One-way ANOVA; n = 4.

(1.04 ± 0.04 Hz for control; 0.45 ± 0.04 Hz for ACTA1-$Ctnnb1^{-/-}$, p < 0.001; n = 4), but no change in mEPP amplitude (*Figure 3A,C*). These results were in agreement with morphological presynaptic deficits (*Figures 1, 2*). The frequency reduction phenotype was rescued by both wild-type Ctnnb1 and the TV-AA mutant (0.95 ± 0.09 Hz in ACTA1-$Ctnnb1^{-/-}$::LSL-*Ctnnb1* and 1.06 ± 0.04 Hz in ACTA1-$Ctnnb1^{-/-}$::LSL-*Ctnnb1*TV-AA, p > 0.05 in comparison with control; n = 4) (*Figure 3A,B*). In contrast, mEPP frequency remained depressed in ACTA1-$Ctnnb1^{-/-}$::LSL-*Ctnnb1*ΔTAD mice (*Figure 3A,B*). Together, these results of both morphological and functional studies indicate the requirement of the Ctnnb1's TAD in motor nerve terminal differentiation and suggest that Ctnnb1-regulated transcription is necessary for motor nerve terminal differentiation.

To determine whether Ctnnb1 is locally activated, we performed the following two experiments (*Figure 2—figure supplement 1*). First, whole-mount staining of muscle fibers with Ctnnb1 antibody indicates that Ctnnb1 in synaptic nuclei appeared to be higher than that in extra-synaptic regions (*Figure 2—figure supplement 1A*). Second, we characterized *Axin2*-nlacZ mice, where the nuclear-localized β-galactosidase (nlacZ) DNA was inserted in the *Axin2* gene, a target of Wnt/Ctnnb1 signaling. Expression of β-galactosidase in *Axin2*-nlacZ mice is controlled, which has been used as a reporter of Wnt or Ctnnb1 activity (*Lustig et al., 2002*). As shown in *Figure 2—figure supplement 1B*, β-galactosidase was detectable in synaptic nuclei, not in extra-synaptic nuclei. These results are in agreement with the notion of active Wnt/Ctnnb1 signaling in synaptic nuclei at the NMJ and provide additional support to the model.

## Identification of Slit2 as a retrograde target

The necessity of the TAD in muscle Ctnnb1 regulation of presynaptic terminals suggests that genes whose expression is controlled by Ctnnb1 may serve as retrograde factors. To identify such a factor, we performed quantitative real-time PCR (qRT-PCR) to screen for genes that are down-regulated in ACTA1-$Ctnnb1^{-/-}$ muscles, compared with $Ctnnb1^{fl/fl}$ control. We focused on secretable and transmembrane proteins in particular muscle-derived extracellular matrix proteins, trophic factors, morphogens, and environmental cues (*Figure 4B–E*). C-Myc and Axin2, whose expression is regulated by Ctnnb1 (*He et al., 1998*; *Yan et al., 2001*), were used a positive control. Their mRNAs were lower in ACTA1-$Ctnnb1^{-/-}$ muscles than control, indicating the validity of the method (*Figure 4A*). In a panel of more than 70 genes, 9 were reduced in expression in ACTA1-$Ctnnb1^{-/-}$, compared to control $Ctnnb1^{fl/fl}$ mice (*Figure 4*). Of particular interest is *Slit2*, a repulsive axon guidance cue of the *Slit* family (*Brose et al., 1999*; *Kidd et al., 1999*). Slit2 as well as Slit1 regulate axon collateral formation (*Nguyen Ba-Charvet et al., 1999*). Interestingly, mutation of *Slit2*, but not *Slit1* or *Slit3*, disrupts motoneuron fasciculation and primary nerve location (*Jaworski and Tessier-Lavigne, 2012*), akin to NMJ morphological deficits of ACTA1-$Ctnnb1^{-/-}$ mice. Of the three members of the *Slit* family, only *Slit2* was down-regulated in ACTA1-$Ctnnb1^{-/-}$ muscles (*Figure 4E*). The reduction was also confirmed by Western blot analysis of ACTA1-$Ctnnb1^{-/-}$ muscle homogenates (*Figure 5A,B*).

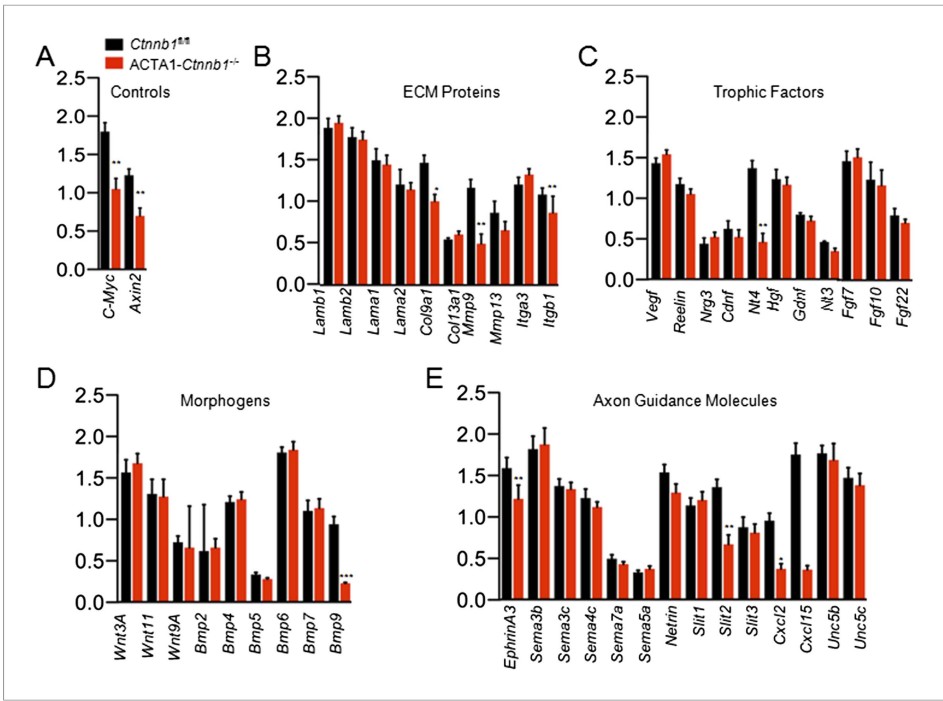

**Figure 4**. *Slit2* was reduced in ACTA1-*Ctnnb1*^−/− muscle. RNA was purified from P0 muscles of *Ctnnb1*^fl/fl (black) and ACTA1-*Ctnnb1*^−/− (red) and subjected to quantitative real-time PCR. Of 70 genes, the expression of 25 was negligible. Shown were genes in different groups: known Ctnnb1 targets (**A**), ECM (extracellular matrix) proteins (**B**), trophic factors (**C**), morphogens (**D**), and environmental cues (**E**). Data are shown as mean ± SEM; *, $p < 0.05$; **, $p < 0.01$; ***, $p < 0.001$; t-test; n = 3.

Muscle homogenates contain, in addition to muscle, nerve terminals, Schwann cells, and blood vessels and connective tissues where Slit2 may be expressed (*Huang et al., 2011*; *Jaworski and Tessier-Lavigne, 2012*). To demonstrate that Slit2 is expressed in muscle cells and the expression is regulated by Ctnnb1, we transfected C2C12 myoblasts with HA-tagged *Ctnnb1* and resulting myotubes were subjected to Western blotting for endogenous Slit2. As shown in *Figure 5C*, elevating Ctnnb1 level increased the expression of endogenous Slit2 in a dose-dependent manner. This result indicates that Ctnnb1 stimulates the expression of Slit2 in muscle cells.

Ctnnb1 regulates gene expression by binding to the *Tcf4/Lef1* transcription factors (*Behrens et al., 1996*; *van de Wetering et al., 1997*). We identified several putative *Tcf4/Lef1*-binding sites in the 5′ UTR of the *Slit2* gene using ChIP-MAPPER (*Marinescu et al., 2005*). There were a *Tcf4* and a *Lef1* site between −1061 and −929 bases and two *Tcf4* sites between −582 and −567 bases upstream from the transcription initiation site (*Figure 5D*). To determine whether Ctnnb1 in fact binds to these sites, we performed chromatin immunoprecipitation (ChIP) assay using C2C12 myotubes. DNA fragments associated with Ctnnb1 were immunoprecipitated with anti-Ctnnb1 antibody and subjected to PCR using primers that flank these two regions (*Figure 5D*). As shown in *Figure 5E*, a specific band was detected using the primers (−582–567), suggesting that Ctnnb1 is associated with the *Tcf4* sites. In contrast, no band was detected using the primers (−1061 and −929), although the primers were able to detect a band in input, suggesting that the *Tcf4* and *Lef1* sites in this region may not be able to bind to Ctnnb1. Together, these results suggest that Ctnnb1 binds to the promoter region of the *Slit2* gene to promote its expression in muscle cells.

### Slit2 overexpression rescues presynaptic defects in ACTA1-*Ctnnb1*^−/− mice

Having demonstrated that *Slit2* is a target of Ctnnb1-regulated transcription in muscles, we wanted to determine if overexpressing Slit2 in muscles was able to rescue presynaptic deficits in ACTA1-*Ctnnb1*^−/− mice. It should if the hypothesis is correct. To this end, we generated a transgenic mouse, ACTA1-*Slit2*, which expressed Flag-Slit2 under the control of the ACTA1 promoter

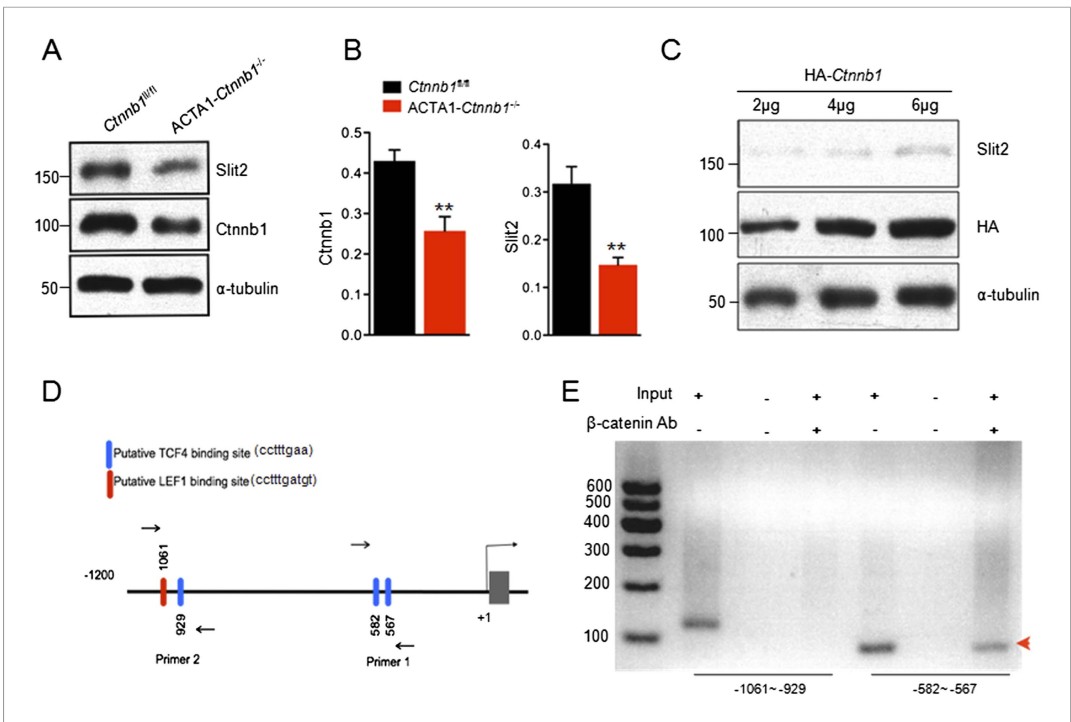

**Figure 5**. Transcriptional regulation of *Slit2* by Ctnnb1. (**A**) Expression of Slit2 was reduced in ACTA1-*Ctnnb1*−/− muscles. (**B**) Quantitative analysis of data in **A**. Data are shown as mean ± SEM; **, p < 0.01; t-test; n = 3. (**C**) Slit2 expression was increased by elevating Ctnnb1. C2C12 myoblasts were transfected with HA-*Ctnnb1*. Resulting myotubes were subjected to Western blot with antibodies against Slit2 or HA. (**D**) Diagram showing putative TCF4/LEF1 binding sites in 5′-UTR. Sites were identified by ChIP-MAPPER (*Marinescu et al., 2005*). Primers for CHIP analysis were indicated by arrows. (**E**) Ctnnb1 binding to the promoter of *Slit2* 5′UTR (red arrowhead). Cross-linked DNA of C2C12 myotubes was subjected precipitation with anti-Ctnnb1 antibody. The complex was used as a template for PCR with the primers indicated in **D**.

(*Figure 6—figure supplement 1A,B*). The ACTA1 promoter drives muscle-specific expression at embryonic day 9.5 (E9.5) (*Crawford et al., 2000*) and has been useful to study NMJ assembly (*Kim and Burden, 2008*; *Wu et al., 2012a*, *2012b*). The transgene Flag-*Slit2* was expressed in muscles of ACTA1-*Slit2* mice, but not muscles of control mice (*Figure 6—figure supplement 1B*). Flag-Slit2 expression was muscle-specific and not detectable in other tissues including the spinal cord (data not shown).

Interestingly, many of presynaptic morphological deficits of ACTA1-*Ctnnb1*−/− mice were rescued by Slit2, including reduced nerve terminal coverage (as indicated by synaptophysin staining) (*Figure 6A–C*), reduced vesicle density (*Figure 7A,B*), and reduced active zones (*Figure 7A,C*). While the number of nerve terminals and synaptic cleft width in either ACTA1-*Ctnnb1*−/− mutant or ACTA1-*Ctnnb1*−/−::ACTA1-*Slit2* rescue mice are not affected (*Figure 7D,E*).These results indicate that overexpressing Slit2 in muscles was sufficient to diminish presynaptic structural deficits that are caused by muscle Ctnnb1 deficiency, in agreement with the notion that *Slit2* acts downstream of Ctnnb1. To test this hypothesis further, we characterized neuromuscular transmission of ACTA1-*Ctnnb1*−/−::ACTA1-*Slit2* mice. Loss of Ctnnb1 in muscles resulted in reduced mEPP frequency but not amplitude (*Figure 7F–H*) (*Li et al., 2008*). Remarkably, mEPP frequency in ACTA1-*Ctnnb1*−/−::ACTA1-*Slit2* mice was higher than that of ACTA1-*Ctnnb1*−/− mice, suggesting again that the vesicle release deficit was rescued by muscle expression of Slit2. The rescue effect of nerve terminal structure and function appeared to be specific because overexpression of Slit2 in muscles had no effect on mislocation of primary nerve branches, reduced number of secondary branches, and increased length of secondary branches (*Figure 6—figure supplement 2A–E*). The effects of muscle expression of Slit2 on post-synaptic deficits were mixed. Ablation of Ctnnb1 in muscles (in ACTA1-*Ctnnb1*−/− mice)

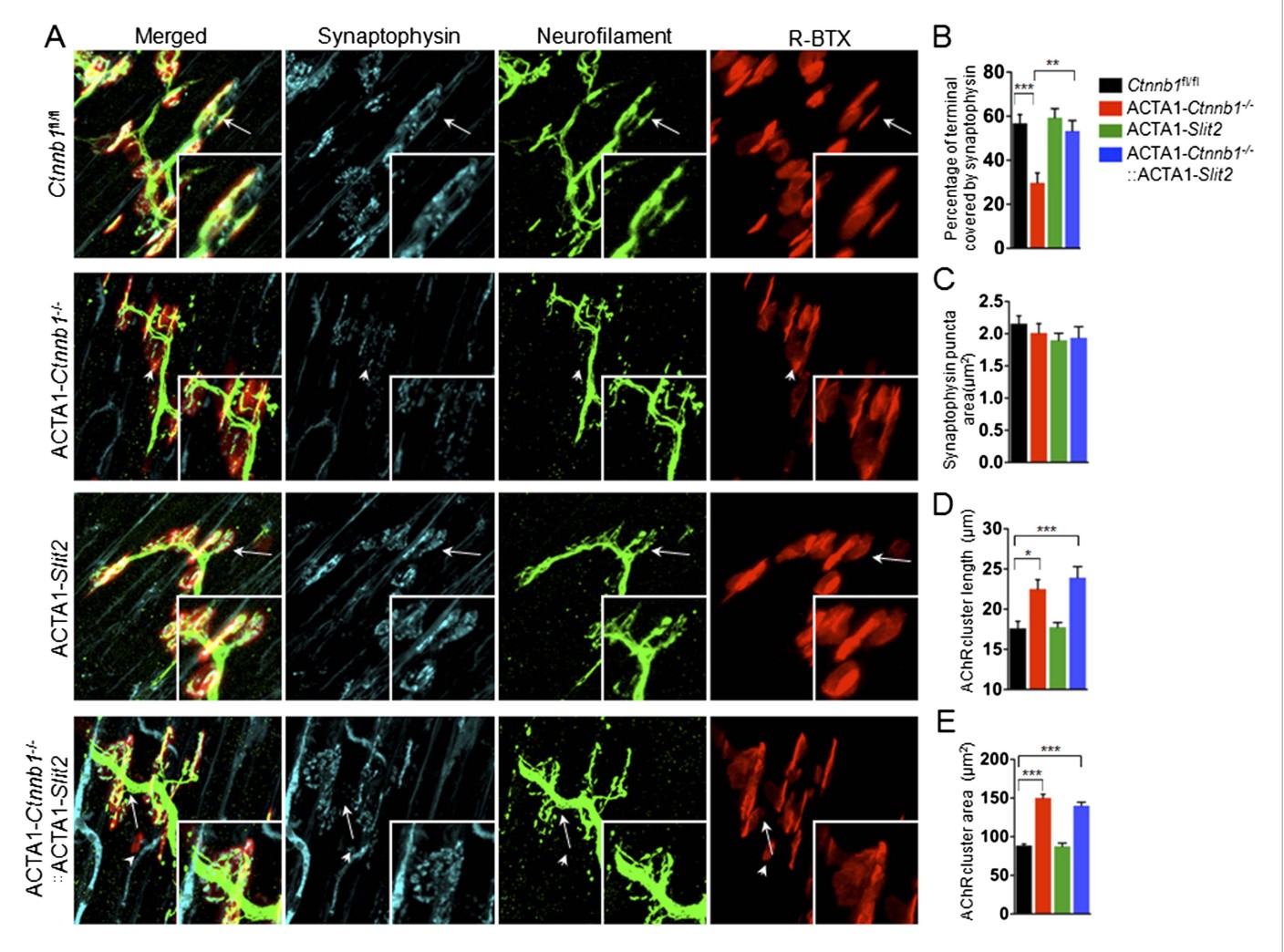

**Figure 6**. Slit2 overexpression increased synaptophysin staining at the NMJ. P0 diaphragms of indicated genotypes were stained whole mount with R-BTX (red) and antibodies against neurofilament to label axons (green), and anti-synaptophysin antibody to label synaptic vesicles (cyan). (**A**) Representative images. Arrows indicate NMJs with synaptophysin; arrowheads indicate NMJ with reduced synaptophysin. (**B–E**) Quantitative analysis of data in **A**. Data are shown as mean ± SEM; *, p < 0.05; **, p < 0.01; ***, p < 0.001; n = 9; One-way ANOVA; bar = 10 µm. NMJ, neuromuscular junction.

The following figure supplements are available for figure 6:

**Figure supplement 1**. Generation of ACTA1-*Slit2* transgenic mice.

**Figure supplement 2**. ACTA1-*Slit2* partially rescues ACTA1-*Ctnnb1*$^{-/-}$ axon arborization defects.

enlarges AChR clusters and increased the central band area where AChR clusters are distributed (*Li et al., 2008*). Slit2 muscle expression had no effect on AChR cluster length or area of ACTA1-*Ctnnb1*$^{-/-}$ mice (*Figure 6A,D,E*). However, the enlarged central band area where AChR clusters are distributed in ACTA1-*Ctnnb1*$^{-/-}$ mice was reduced (*Figure 6—figure supplement 2A,E*). Together, these observations indicate muscle-specific expression of Slit2 was able to rescue presynaptic structural and functional deficits at nerve terminals.

## Slit2 promotion of terminal differentiation of motoneurons

The finding that muscle expression of Slit2 is sufficient to increase vesicle numbers and active zones in Ctnnb1-deficient mice suggested to us that Slit2 may be synaptogenic. Once released from muscles, Slit2 may be concentrated at the NMJ, like muscle-expressed AChE

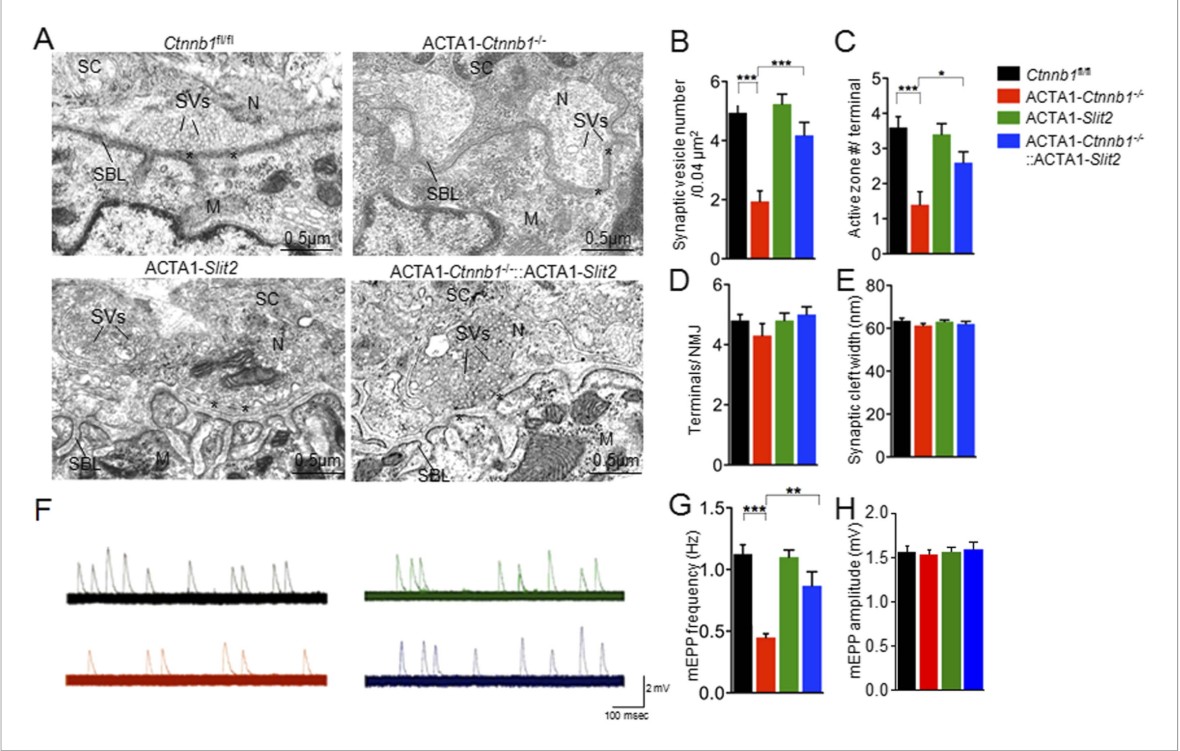

**Figure 7**. Rescue of nerve terminal deficits by ACTA1-*Slit2*. (**A**) Rescue of nerve terminal morphology. Representative electron micrographic images of indicated genotypes. Asterisks, active zones; SBL, synaptic basal lamina; N, nerve terminals; M, muscle fibers; SC, Schwann cells; SVs, synaptic vesicles. (**B–E**) Quantitative analysis of data in **A**. (**F**) Rescue of mEPP frequency reduction. Shown were representative mEPP traces. (**G–H**) Quantitative analysis of mEPP amplitude and frequency, respectively. Data were shown as mean ± SEM; *, p < 0.05; **, p < 0.01; ***, p < 0.001; One-way ANOVA; n = 10 for **B–E**; n = 4 for **G–H**.

(*Rotundo et al., 2008*; *Barik et al., 2014a*). To test these hypotheses, we stained muscles with anti-Slit2 antibody. As shown in *Figure 8A*, the immunoreactivity co-localized with R-BTX and synaptophysin (*Figure 8A*), indicating that Slit2 is enriched in synaptic basal lamina (SBL). To determine whether concentrated Slit2 was able to induce nerve terminal differentiation, we expressed Slit2 in HEK293 cells (*Figure 8B*). Purified Slit2 was immobilized on fluorescently conjugated latex microspheres (beads), which were then incubated with spinal cord explant (ventral horn) in culture (*Figure 8C*). The culture was stained for synaptophysin 24 hr later. As shown in *Figure 8C–E*, 46.9 ± 2.62% of Slit2-beads were positive for synaptophysin, whereas, in contrast, only 18.6 ± 1.47% of controls were synaptophysin-positive (n = 10, p < 0.001) (*Figure 8C,D*). Moreover, the size of synaptophysin puncta was also increased with Slit2-conjugated beads, compared with controls (*Figure 8C,E*). Together, these results suggest that immobilized Slit2 may be sufficient to induce synaptophysin clusters of cultured spinal cord explant.

## Discussion

Many molecules have been identified to regulate presynaptic differentiation at the NMJ. For example, muscle-derived FGF 7/10/22 and collagens IV and XIII were shown to induce synaptic vesicle clusters at nerve terminals or regulate presynaptic maturation (*Fox et al., 2007*; *Latvanlehto et al., 2010*). In *Drosophila*, wishful thinking and Glass bottom boat, a TGFβ receptor and ligand, respectively, were shown to be important for presynaptic T-bar formation (*Marques et al., 2000*; *McCabe et al., 2003*, *2004*). GDNF (glial cell derived neurotrophic factor), known to affect motoneuron survival in vitro, was shown to affect pre- and post-synaptic neurotransmission in frog neuron-muscle co-cultures (*Oppenheim et al., 1995*; *Keller-Peck et al., 2001*; *Wang et al., 2002*). Laminin β2, a protein concentrated in SBL, could inhibit neurite outgrowth and regulate nerve terminal differentiation and maturation (*Noakes et al., 1995*; *Fox et al., 2007*). It is thought to bind directly to and cluster calcium

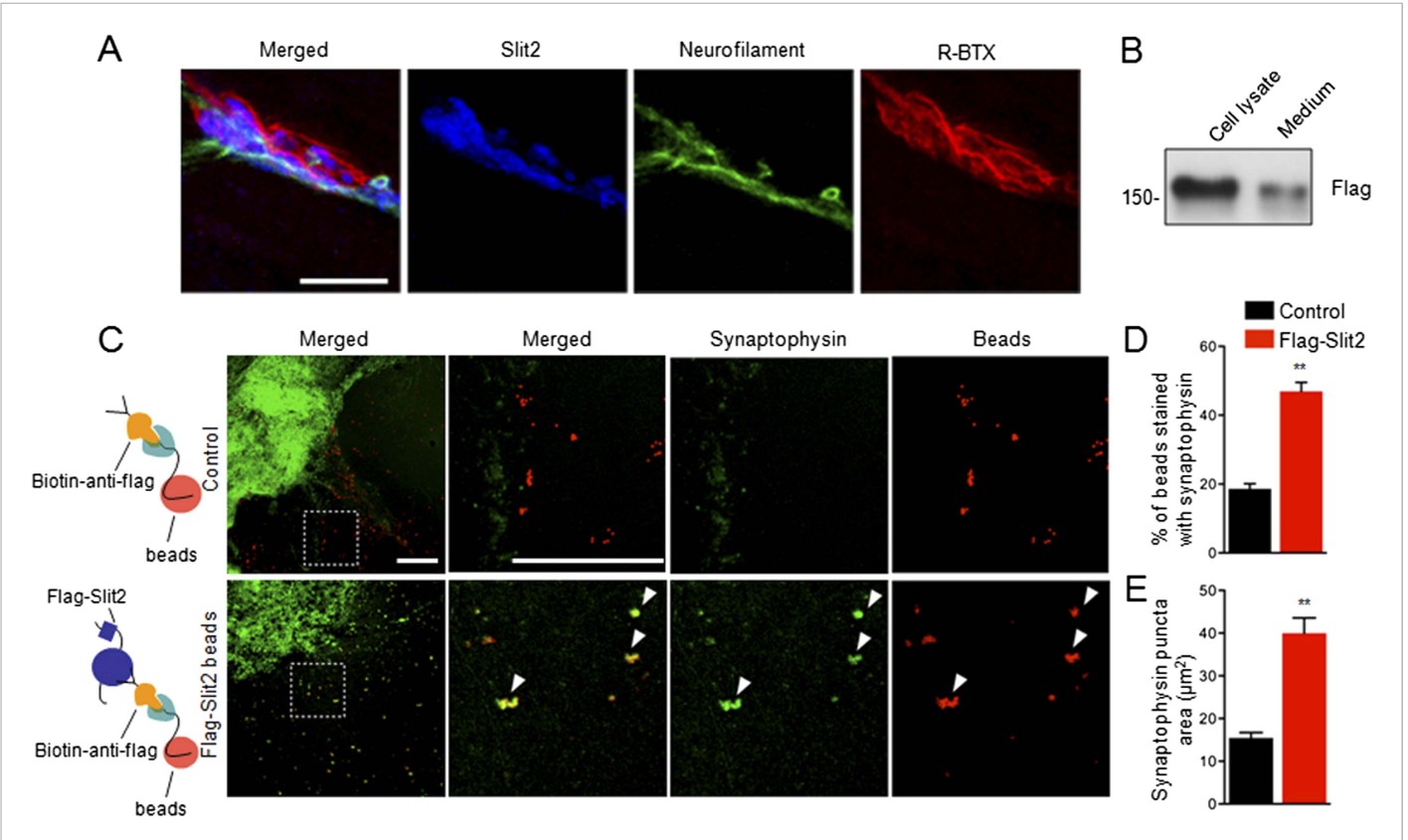

**Figure 8**. Slit2 was enriched at the NMJ and induced synaptophysin puncta of spinal cord explants. (**A**) P12 tibilalis anterior muscles stained with R-BTX (red), anti-neurofilament antibody (green), and anti-Slit2 antibody (blue). (**B**) Detection of Flag-Slit2 in condition medium of transfected HEK293 cells. (**C**) Increased synaptophysin puncta co-localization with Flag-Slit2-conjugated beads. (**D–E**) Quantitative analysis of data in **C**, Data are shown as mean ± SEM; **, $p < 0.01$; t-test; bar = 10 µm for **A**; bar = 500 µm for **C**.

channels, which in turn recruit other presynaptic components (*Nishimune et al., 2004*). In Integrinβ1 muscle-specific mutant mice, nerve terminal arborization and differentiation are abnormal (*Schwander et al., 2004*). Lrp4 and MuSK were shown to be critical for presynaptic differentiation independently (*Kim and Burden, 2008*; *Yumoto et al., 2012*; *Wu et al., 2012b*).

Previously, we demonstrated that Ctnnb1 in muscle is critical for the differentiation of motor nerve terminals (*Li et al., 2008*). In this paper, we dissected out which functions of muscle Ctnnb1 is critical by an in vivo transgenic approach. We showed that muscle overexpression of the Ctnnb1 mutant without the TAD was unable to rescue the presynaptic deficits of Ctnnb1 mutation, indicating that the transcription regulation is critical for muscle Ctnnb1 to control presynaptic differentiation. On the other hand, the cell-adhesion function of Ctnnb1 is dispensable. These results suggest that muscle Ctnnb1 is likely to regulate presynaptic differentiation by controlling the expression of muscle-derived proteins. This notion was supported by the observations that Ctnnb1 and Wnt signaling reporter were higher in synaptic nuclei at the NMJ than those in extra-synaptic areas. We reasoned that such a protein is secretable or has a transmembrane domain and its expression is down-regulated in Ctnnb1 mutant muscle. A screen for proteins that fulfill these criteria led to the identification of *Slit2*. We showed, first, that Ctnnb1 associates with *Tcf/Lef* cis-elements in the promoter region of the *Slit2* gene. Second, *Slit2* expression in muscle cells is promoted by elevating Ctnnb1. Slit2 was concentrated at the NMJ. Third, transgenic mice expressing Slit2 specifically in the muscle were able to diminish presynaptic deficits by Ctnnb1 mutation, including poor innervation, reduced vesicle density and active zone, and depressed mEPP. Finally, Slit2 immobilized on beads was able to induce the synaptophysin puncta of axons of spinal cord explants. Together, these observations suggest that Slit2 serves as a factor utilized by muscle Ctnnb1 to direct presynaptic differentiation.

Slit2 is a member of the Slit family of secreted glycoproteins, which serve as repulsive cues for axons of various neurons (*Nguyen-Ba-Charvet et al., 2002*; *Shu et al., 2003*; *Marillat et al., 2004*; *Sabatier et al., 2004*; *Hussain et al., 2006*), as regulators to promote axonal and dendritic branching (*Brose et al., 1999*; *Wang et al., 1999*; *Nguyen Ba-Charvet et al., 2001*; *Whitford et al., 2002*; *Miyashita et al., 2004*), and as repellants for neuron migration (*Wu et al., 1999*). A previous work showed that motor neurons use Slit2 to promote their axon fasciculation by an autocrine or juxtaparacrine mechanism (*Jaworski and Tessier-Lavigne, 2012*). In *Slit2* mutant mice, motor nerves prematurely defasciculate into smaller bundlers in various muscles, unlike those in wild type that stay in compact fascicle until they reach the middle of muscle fibers. NMJ morphology at gross anatomic level appears to be normal in *Slit2* mutant mice (*Jaworski and Tessier-Lavigne, 2012*). We show that Slit2 is produced by muscle cells in a manner dependent on Ctnnb1. Remarkably, Slit2, when immobilized on beads, was synaptogenic—to promote the formation and enlargement of synaptophysin-positive punta. Soluble Slit2 did not have this effect, suggesting that the effect requires local, high concentration as many synaptogenic proteins do (*Dean et al., 2003*; *Yumoto et al., 2012*). This high concentration in vivo may be achieved by its enrichment at the NMJ, probably due to the involvement of the laminin-G domain of Slit2's extracellular region. The laminin-G domain is known to mediate the SBL immobilization of many proteins including agrin, α-dystroglycan, perlecan, and neurexin (*Timpl et al., 2000*; *Chedotal, 2007*). The molecular mechanism of Slit2's synaptogenic effect warrants future investigation. Interestingly, the extracellular region of Slit2 has four leucine-rich repeats (LRR)-11-residue sequence rich in leucines at defined positions (LxxLxLxxNxL, x being any amino acid). The LRRs have been conserved in some synaptogenic proteins (*de Wit et al., 2011*).

The presynaptic deficits in ACTA1-*Ctnnb1*$^{-/-}$ mice can be grouped into two. One includes mislocation of the primary branches of the phrenic nerve and secondary branch arborization and length, while the other consists of synaptic vesicle number reduction, fewer active zones, and reduced mEPP frequency that was due to reduced release probability (*Li et al., 2008*). This observation suggests that the 'trophic' activity of limb buds or muscles identified in the classic limb bud removal experiment is mediated by a Ctnnb1-independent mechanism. Interestingly, Slit2 muscle expression was able to rescue the second group of presynaptic deficits. Besides *Slit2*, several other secretable factors were found lower in ACTA1-*Ctnnb1*$^{-/-}$ muscles. Their function in NMJ formation may be limited or opposite of the hypothetic factor downstream of muscle Ctnnb1. For example, *Mmp9* and *Cxcl15* mutant mice are viable and presumably have normal NMJ function (*D'Arcangelo et al., 1996*; *Coussens et al., 2000*; *Chen et al., 2001*), excluding their involvement in the Ctnnb1 regulation. *Hgf* was shown to be necessary for motor neuron survival (*Wong et al., 1997*; *Yamamoto et al., 1997*); however, the number of motor neurons was not reduced in ACTA1-*Ctnnb1*$^{-/-}$ mice or in mice overexpressing Ctnnb1 in muscles(*Wu et al., 2012a*). β1-Integrin in muscle is known to be required for NMJ formation (*Schwander et al., 2004*). In particular, axons in β1-Integrin mutant mice failed to respect muscle boundaries and grew onto the tendon organ. However, they do not have nerve defasciculation defects as observed in Ctnnb1-mutant mice. These results do not support a possible involvement of these factors in muscle Ctnnb1 regulation of presynaptic differentiation. It is worthy pointing out that Slit2 overexpression in muscles has no effect on deficits involving primary and secondary branches. Therefore, presynaptic differentiation may be regulated by multiple factors derived from the skeletal muscle.

Recent evidence suggests that Wnt signaling regulates NMJ formation. Some Wnts can stimulate while others inhibit agrin-induced AChR clustering (*Henriquez et al., 2008*; *Strochlic et al., 2012*; *Zhang et al., 2012*; *Barik et al., 2014b*). The receptor of Wnts at the NMJ may include MuSK that has a CRD domain homologous to that in the Wnt receptor Frizzled (*Masiakowski and Yancopoulos, 1998*) and LRP4, a member of the LDLR family whose members, like LRP5/6, serve are Wnt receptors (*Tamai et al., 2000*). In zebrafish, MuSK is thought to interact with Wnt11r to regulate NMJ formation (*Zhang et al., 2004*; *Jing et al., 2009*). Moreover, many Wnt signaling molecules have been implicated in NMJ development, including Dvl and APC (*Luo et al., 2002*; *Wang et al, 2003*). The deletion of the CRD domain of MuSK in mice causes both pre- and post-synaptic defects of the neuromuscular junction (*Messéant et al., 2015*). Whether Ctnnb1 regulation of presynaptic differentiation is downstream of Wnts is unclear. There are 19 Wnt ligands in mice, which makes it difficult to investigate their function by mutant mice. Perhaps due to functional redundancy, mice lacking Wnt4 or Wnt11, an isoform most homologous to Wnt11r in zebrafish, were able to form NMJ (*Banerjee et al., 2011*; *Strochlic et al., 2012*).

# Materials and methods

## Mice strains

*Ctnnb1* conditional KO mice were generated by crossing floxed *Ctnnb1* (*Ctnnb1*<sup>fl/fl</sup>) (Jackson Laboratory; stock #:004152) with ACTA1-Cre transgenic mice that were generously provided by Dr. Melki, to generate the ACTA1-C*tnnb1*$^{-/-}$ mice which lacked Ctnnb1 in skeletal muscles. For the rescue experiments, three loxP-STOP-loxP (LSL)-*Ctnnb1* transgenic mice (wild-type *Ctnnb1*; *Ctnnb1*TV-AA and *Ctnnb1*ΔTAD) were generated. *Ctnnb1*TV-AA and *Ctnnb1*ΔTAD (lacking amino acids 684–781) were generated using PCR. The HA tag at the C-terminus was added by subcloning the transgenes into the SalI and XbaI sites of pKH3. HA-tagged wild type and mutants were subcloned into the XhoI and NotI sites of pCCALL2 (loxP-STOP-loxP) vector to generate LSL-*Ctnnb1* or mutant mice. LSL-transgenic mice were crossed with *Ctnnb1*<sup>fl/+</sup> to generate *Ctnnb1*<sup>fl/+</sup>;LSL-*Ctnnb1* or mutant mice. To generate ACTA1-*Slit2* mice, Flag-*Slit2* was subcloned into pBSX-ACTA1 as described previously (*Monks et al., 2007*). *Axin2*-nlacZ mice were obtained from Jackson laboratory (stock #:009120). DNA sequences of all transgenes were validated by sequencing. Linearized DNA was microinjected into FVB blastocysts, which were introduced into pseudo-pregnant recipient females. Mice were genotyped by PCR analysis of tail biopsy DNA. Tail DNA was digested with buffer A (25 mM NaOH, 0.2 mM EDTA (ethylenediaminetetraacetic acid)) for 45 min at 100°C and neutralized with buffer B (40 mM Tris–HCl, pH 8.0) for genotyping. Mice were housed in cages that were maintained in a room with 12-hr light/dark cycle with ad libitum access to water and rodent chow diet (Diet 1/4″ 7097, Harlan Teklad). P0 pups of either sex were analyzed, unless otherwise indicated. 'Material and method' have been approved by the Institutional Animal Care and Use Committee (IACUC) of Georgia Regents University.

## Western blot analysis

Muscles were lysed in modified RIPA buffer (50 mM Tris–HCl, pH 7.4, 150 mM NaCl, 1% NP-40, 2% sodium dodecyl sulfate (SDS), 2% deoxycholate, 1 mM PMSF (phenylmethanesulfonylfluoride), 1 mM EDTA, 5 mM sodium fluoride, 2 mM sodium orthovanadate, and protease inhibitors including 1 mM phenylmethylsulfonyl fluoride, 1 μg/μl pepstatin, 1 μg/μl leupeptin, and 2 μg/ml aprotinin). After centrifuging at 10,000 RPM at 4°C, supernatant was designated as lysates. Protein concentration was measured using Pierce BCA kit. Samples (50 μg of protein) were resolved by SDS-PAGE and transferred to nitrocellulose membrane, which was incubated in 5% milk in phosphate buffer saline (PBS)-0.3% Tween20 overnight at 4°C and then with primary antibodies in 2% milk in PBS-Tween buffer: Ctnnb1 (1:2000, Cell Signaling #9562) (Danvers, MA); Slit2 (1:1500, Abcam, ab7665) (San Francisco, CA); HA (1:2000, Sigma, H9658) (St. Louis, MO); Flag (1:1500, Sigma, F3165); GFP (1:1000, Sigma, G1546); Cre (1: 400, Abcam, Ab24607); β-actin (1:4000, Sigma, A5441). After washing, the membrane was incubated with PBS-Tween buffer containing HRP-conjugated goat anti-mouse and rabbit IgG from Pierce (1:5,000, PI-31430 [anti-mouse], PI-31460 [anti-rabbit]). Immunoreactive bands were visualized by using enhanced chemiluminescence (Life Technologies, Grand Island, NY). Quantitative densitometry of captured images was analyzed with Image J (NIH), as described before (*Barik et al., 2014a*).

## Light microscopic analysis

Diaphragms were stained whole mount as previously described with modification (*Li et al., 2008*; *Wu et al., 2012a*, *2012b*). Briefly, entire diaphragms with ribs were freshly dissected from mice and fixed in 4% paraformaldehyde (PFA) in PBS (pH7.3) at 4°C for 3 days, rinsed with PBS (pH 7.3) at 25°C, and incubated with 0.1 M glycine in PBS for 1 hr at room temperature. The pH of PBS including PFA was critical for proper staining. After rinse with 0.5% Triton X-100 in PBS, diaphragms were incubated in the blocking buffer (5% bovine serum albumin [BSA] and 1% Triton X-100 in PBS) for 3 hr at room temperature. Tissues were then incubated with primary antibodies in the blocking buffer overnight at room temperature on a rotating shaker. After washing 3 times for 1 hr each with 0.5% Triton X-100 in PBS, tissues were incubated with Alexa Fluor 488 or Alexa Fluor 633-conjugated antibody to rabbit or mouse IgG (1:500, Invitrogen, Carlsbad, CA) and rhodamine-conjugated α-BTX/alexa-488 conjugated α-BTX (1:2000, Invitrogen, Carlsbad, CA) overnight at room temperature. After washing 3 times for 1 hr each with 0.5% Triton X-100 in PBS and rinsing once with PBS, tissues were flat-mounted in Vectashield mounting medium (H-1000, Vector laboratories, Burlingame, CA). Z-serial images were

collected with a Zeiss confocal laser scanning microscope (LSM 510 META 3.2) and collapsed into a single image. For quantitative analysis, images were captured at a subsaturating amplifier gain without modification and quantified using LSM Image Browser and FIJI (NIH) software (*Wu et al., 2012a*, *2012b*; *Barik et al., 2014a*).

## Electron microscopy analysis

Electron microscopy was performed as described previously (*Wu et al., 2012a*, *2012b*; *Barik et al., 2014a*). Entire diaphragms were lightly stained with R-BTX (1:1000, in ice cold PBS at 4°C) to mark the central region where NMJs are enriched. The regions were dissected with a micro scalpel (Harvard apparatus, #PY2 56-5673) under a Leica fluorescent dissection scope and fixed (blocks of approximately 4 mm × 4 mm) in 2% glutaraldehyde and 2% PFA in 0.1 M PBS for overnight at 4°C. Tissues were further fixed in sodium cacodylate-buffered (pH 7.3) 1% osmium tetroxide for 1 hr at 25°C. Fixed tissues were washed 3 times with PBS and subjected to dehydration through a series of ethanol: 30%, 50%, 70%, 80%, 90%, and 100%. After 3 rinses with 100% propylene oxide, samples were embedded in plastic resin (EM-bed 812, EM Sciences). Serial sections (1–2 μm) were stained with 1% Toluidine Blue to identify motor nerves and were cut into ultra-thin sections. Alternate longitudinal sections were not chosen to avoid duplicity of obtaining images from same terminals. They were mounted on 200-mesh unsupported copper grids and stained with a solution containing 3% uranyl acetate, 50% methanol, 2.6% lead nitrate, and 3.5% sodium citrate (pH 12.0). Electron micrographs were taken using a JEOL 100CXII operated at 80 KeV.

## Electrophysiological recording

Recording was performed as described previously (*Li et al., 2008*; *Wu et al., 2012a*, *2012b*; *Barik et al., 2014a*). Neonatal mice diaphragms with ribs and intact phrenic nerves were dissected in oxygenated (95% $O_2$/5% $CO_2$) Ringer's solution (136.8 mM NaCl, 5 mM KCl, 12 mM $NaHCO_3$, 1 mM $NaH_2PO_4$, 1 mM $MgCl_2$, 2 mM $CaCl_2$, and 11 mM D-glucose, pH 7.3) and pinned on Sylgard gel in a dish perfused with oxygenated Ringer's solution. To measure mEPP, microelectrodes, 20–50 MΩ when filled with 3 M KCl, were pierced into the center of muscle fibers with the resting potential between −45 and −55 mV. To evoke end-plate potentials, phrenic nerves were stimulated by a suction electrode with suprathreshold square pulses (0.1 ms) using Master-8 (A.M.P.I, Jerusalem, Israel). Data were collected with axonpatch 200B amplifier (Molecular Devices, Sunnyvale, CA), digitized with Digidata 1322A (Molecular Devices, Sunnyvale, CA), and analyzed using pClamp 9.2 (Molecular Devices, Sunnyvale, CA).

## Cell culture and transfection

Mouse muscle C2C12 myoblasts were propagated and induced to form myotubes as described previously (*Luo et al., 2003*). C2C12 myoblasts were transfected with lipofectamine 2000 (Invitrogen, 11,668-019) with a modified protocol. C2C12 myoblasts, at 70–80% confluence, were rinsed once with serum-free medium before transfection because serum appeared to reduce transfection efficiency. After complete aspiration of the medium, myoblasts were incubated with a mixture of DNA, lipofectamine, and serum-free medium for 8 hr when the medium was changed to the growth medium. The DNA:lipofectamine ratio in the mixture was 1 μg:2 μl. The optimal volume of the mixture for 24-well dishes was 200 μl medium per well that contained 2 μg plasmid DNA. HEK293 cells were cultured and transfected as previously described (*Luo et al., 2008*; *Wu et al., 2012b*).

## ChIP

ChIP assays were performed according to the protocol provided by Upstate Biotechnology (*Kim et al., 2005*). C2C12 myotubes were cultured on gelatin-coated 6-well dish, transfected with HA-*Ctnnb1*, were washed twice with PBS and then incubated in 1% formaldehyde in PBS at room temperature for 10 min. After washing twice with PBS, cells were scraped in PBS containing protease inhibitors, centrifuged at 2000 rpm at 4°C, and resuspended in SDS lysis buffer containing 1% SDS, 10 mM EDTA, 50 mM Tris–HCl (pH 8.0) plus protease inhibitors. Cells were sonicated by Sonic Dismembrator model 100 (Fisher, Suwanee, GA), strength 3, for 15 s 3 times, to shear chromosome DNA. After centrifugation at 13,000 at 4°C, the sonicated mixture was diluted 10-fold with ChIP dilution buffer containing 0.01% SDS, 1.1% Triton X-100, 1.2 mM EDTA, 16.7 mM Tris–HCl (pH 8.1), and 167 mM NaCl. The mixture was precleaned once with salmon sperm DNA-protein A-agarose beads at 4°C for 1 hr. The resulting supernatant was incubated with anti-Ctnnb1 antibody at 4°C

overnight, when salmon sperm DNA-protein A-agarose beads (50 µg/µl) was added. The beads/mixture was incubated for another hour at 4°C and washed sequentially with a low-salt wash buffer containing 0.1% SDS, 1% Triton X-100, 2 mM EDTA, 20 mM Tris–HCl (pH 8.1), and 150 mM NaCl; a high-salt wash buffer containing 0.1% SDS, 1% Triton X-100, 2 mM EDTA, 20 mM Tris–HCl (pH 8.1), and 500 mM NaCl; an LiCl wash buffer containing 0.25 M LiCl, 1% NP-40, 1% deoxycholic acid, 1 mM EDTA, and 10 mM Tris–HCl (pH 8.1); and a wash with TE buffer. Beads were then incubated twice with 250 µl of the elution buffer (1% SDS, 0.1 M NaHCO3) for 15 min at room temperature to elute bound DNA. Combined eluates were added to 20 µl of 5 M NaCl to reverse cross-links by heating at 65°C for 4 hr. After incubation with 10 µl of 0.5 M EDTA, 20 µl of 1 M Tris–HCl (pH 6.5), and 2 µl of 10 mg/ml proteinase K (1 hr at 45°C), DNA was recovered by phenol-chloroform extraction and ethanol precipitation and used as template for PCR. Primers for the two TCF4/ LEF1-binding regions were as follows; *Slit2* (primer1)- 5′-GTCCC CTTTA GGATC GCG-3′/5′-CAGCG GGAGA ACGAG G-3′ and *Slit2* (primer2)-5′-AGAGA AGGTT GAAAA CACTA CTCCC-3′/5′-TGAAA TAGAT CTGCC TCCGT G-3′.

## qRT-PCR

Total RNA was isolated from muscles using Trizol (Invitrogen) and purified using the RNeasy mini kit (Qiagen). Quantitative RT-PCR was performed as described previously (*Wu et al., 2012a*). Equal amounts of total RNA (1 µg) were reverse transcribed by random hexamer primers using the Maxima Enzyme Mix (Fermentas). Quantitative PCRs were run in a PTC-200 Peltier Thermal Cycler (Bio-Rad MJ Research) using SYBR Green/ROX (Fermentas). The primers for specific genes are listed in *Supplementary file 1*.

## Synaptogenic assay

Spinal cord explant culture from E12.5 embryos was performed as previously described (*Wang and Marquardt, 2012*). In short, pregnant mouse was euthanized at E12.5 after anesthesia by isoflurane and subsequent cervical dislocation. Embryos along with amniotic sacs were dissected and placed in ice-cold PBS. Spinal cords were dissected out of the embryos exposing ventral motoneuron columns. Ventral horns were dissected out by a scalpel and cultured on laminin (Invitrogen #23017-15) coated coverslips in neurobasal medium (catalog #21103-049; Invitrogen) supplemented with B27 (catalog #17504-044; Invitrogen), 600 µM/ml-glutamine (catalog #25- 005-CI; Cellgro), BDNF (2 ng ml$^{-1}$), GDNF (2 ng ml$^{-1}$), CTNF (2 ng ml$^{-1}$), NGF (1 ng ml$^{-1}$) (Sigma), and penicillin-streptomycin (catalog #30-003-CI; Cellgro). Every alternate day, half of the medium was replaced by freshly prepared medium. 6 days after plating, the explants are subjected to incubating with Slit2-conjugated beads (see below).

HEK293 cells were transfected with Flag-*Slit2* or control empty Flag vector, to produce Flag-Slit2 and control conditioned media, respectively. Biotin-conjugated mouse monoclonal antibody (2 µl) (anti-Flag BioM2, Sigma, F9291) was conjugated with 50 µl of FluoSpheres NeutrAvidin-labeled Microspheres (1.0 µm, red fluorescent [580/605], F-8775, Invitrogen) by incubation in PBS for 4 hr at 4°C and then washed with ice-cold PBS, 3 times. Beads were re-suspended in 200 µl of ice-cold PBS with protease inhibitors and incubated with Flag-Slit2 or control conditioned media (200 µl) overnight on a rotating shaker at 4°C. Flag-Slit2-conjugated neutravidin microspheres were isolated by spinning at 5000 rpm at 4°C. Microspheres were washed 3 times with ice-cold PBS and diluted in explant culture media (1:100) and incubated with spinal cord explants for 24 hr before analysis.

Dorsal horn explants were fixed 4% ice-cold PFA for 20 min at room temperature and washed twice (5 min each) with 1 ml PBS. Explants were then permeabilized by 1 ml blocking solution (PBS with 0.3% Triton X-100, 5% BSA, and 3% goat serum) for 5 min at room temperature and incubated with primary antibodies in blocking solution for 2 days at 4°C. Explants were rinsed 3 times with 1 ml PBS, 5 min each, and incubated with secondary antibodies in blocking solution for 90 min at room temperature. After washing thrice with 1 × PBS, 5 min each, coverslips were mounted on glass slides and covered with Vectashield mounting medium (H-1000; Vector Laboratories). Images were obtained by confocal scanning microscopy. FIJI (NIH) was used to measure and quantify synaptophysin puncta number and size.

## Statistical analysis

Two-tailed student *t*-test was used to compare data between two groups. One-way ANOVA (analysis of variance) was used to compare data between multiple groups, and Bonferroni's Multiple Comparison Test was used to determine the significance. Unless otherwise indicated, data were expressed as mean ± SEM, and considered statistically significant if p value was less than 0.05.

## Acknowledgements

We are grateful to Dr Jane Wu for valuable Slit2 constructs and Dr Chien-Ping Ko for advice on EM analysis. We thank members of the Mei and Xiong laboratories for discussion and the GRU EM Core for assistance. This work was supported in part by grants from National Institutes of Health (LM and WCX), Muscular Dystrophy Association (LM), and the State Key Development Program for Basic Research of China (973 Program) (2014CB542203), National Natural Science Foundation of China (31240074 and 31371149) and Beijing Nova Programme (Z131102000413033) (HW).

## Additional information

### Funding

| Funder | Grant reference | Author |
| --- | --- | --- |
| Muscular Dystrophy Association (Muscular Dystrophy Association Inc.) | MDA240849 | Lin Mei |
| National Institute of Neurological Disorders and Stroke (NINDS) | NS082007 and NS056415 | Lin Mei |
| National Basic Research Program (973 Program) | 2014CB542203 | Haitao Wu |
| National Natural Science Foundation of China (NSFC) | 31240074, 31371149 | Haitao Wu |
| Beijing Nova Program | Z131102000413033 | Haitao Wu |
| National Institute of Aging | AG045781 | Wen-Cheng Xiong |

The funders had no role in study design, data collection and interpretation, or the decision to submit the work for publication.

### Author contributions

HW, AB, Conception and design, Acquisition of data, Analysis and interpretation of data, Drafting or revising the article; YL, CS, Acquisition of data, Analysis and interpretation of data; AB, TWL, Acquisition of data, Drafting or revising the article; LL, AS, Analysis and interpretation of data, Drafting or revising the article; WX, Conception and design, Drafting or revising the article; LM, Conception and design, Analysis and interpretation of data, Drafting or revising the article

### Author ORCIDs

Arnab Barik, http://orcid.org/0000-0001-6850-0894

### Ethics

Animal experimentation: This study was performed in accordance to experimental procedures and protocols (# 2011-0393) approved by the Institutional Animal Care and Use Committee (IACUC) of the Georgia Regents University. Every effort was made to minimize the suffering of the experimental animals.

## Additional files

### Supplementary file

• Supplementary file 1. The sequences of primers used for qRT-PCR.

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
