## [Decision Letter]

Thank you for sending your work entitled “Slit2 as a β-catenin-dependent retrograde signal for presynaptic differentiation” for consideration at *eLife*. Your article has been favorably evaluated by a Senior editor, a Reviewing editor, and three reviewers.

We include the three reviews at the end of this letter, as there are various specific comments in them that will not be repeated in the summary here.

All of the reviewers were impressed with the importance and novelty of your work. Overall, the experiments appear to be carefully executed. However, there is a general consensus that the data should be more cautiously interpreted and that various experiments that could strengthen (or weaken) a number of the conclusions are feasible.

We would like to encourage you to resubmit a revised manuscript that addresses the specific issues raised in the reviews. We realize that several of the more ambitious suggestions are beyond the scope of the present work, but we have not removed them from the reviews since they convey the reaction to the work overall.

*Reviewer #1*:

In this manuscript, Wu et al. investigated the mechanism by which β-catenin controls presynaptic differentiation. The present work is an extension of their previous work (Nat Neurosci 11: 262-268), in which they showed that expression of β-catenin in muscle is critical for motoneuron differentiation. In this current work, they identified Slit2 as the retrograde factor activated by β-catenin. Specifically, 1) they showed that β-catenin TAD domain is required for rescuing presynaptic deficits of NMJ deficient in β-catenin; 2) they identified Slit2 as the retrograde factor activated by β-catenin; 3) transgenic expression of Slit2 rescued the presynaptic deficits of NMJs in β-catenin mutant mice.

Overall this is a very interesting study which uncovers an unexpected retrograde factor Slit2 controlled by β-catenin. The manuscript is well written, the analyses of data were carefully performed and main conclusions are supported by the provided results. Thus the publication of the paper would be of interest to the general readership of the journal.

*Reviewer #2*:

This paper addresses how the β-catenin protein, earlier implicated in neuro-muscular junction (NMJ) formation, operates at the molecular level. The authors characterize transgenic mice that express wild-type or mutant forms of β-catenin, constructed such that the transcriptional activity and the cell adhesion functions of β-catenin are separated. By mutant rescue experiments, a role for the transcription by β-catenin is revealed. The target gene relevant for the activation of muscle-derived retrograde factors is identified as Slit2, a protein known to be cue for axon guidance but (to the knowledge of this reviewer) not yet implicated in NMJ formation. The Slit2 promoter may have essential binding sites for the β-catenin-TCF complex. There are some experimental issues that should be addressed.

To start with a general question, if Slit2 is a transcription target of the β-catenin-TCF complex, it would be likely that β-catenin is activated by a Wnt signal, as the majority of cases of β-catenin acting as a transcription factor are due to Wnt signals. In the muscle, this would be interesting to know, in particular where the Wnts are expressed. In Figure 4, some Wnts are tested for overall expression but this is a limited survey. As a locally acting signal, the Wnt protein could contribute, in a deterministic way, where on the muscle fiber β-catenin and subsequently Slit2 would be expressed, setting up the site of the NMJ. It could of course be that Wnt is coming from the incoming neuron. I realize that possible local signaling in a muscle fiber is complicated by the multinucleated nature of the muscle, but it still remains an important question whether β-catenin is locally activated. The latter point could be examined by a simple stain for β-catenin in the muscle while the use of Wnt/TCF reporter mice could indicated local transcriptional activation.

The survey of secreted molecules (Figure 4) shows that apart from Slit2, several others seem to be regulated by the absence of β-catenin as well. It is not clear why these genes were excluded from further analysis.

Regarding the experiments that show a role of β-catenin-TCF in the activation of Slit2, the text mentions binding sites in the 5'UTR (subsection “Identification of Slit2 as a retrograde target”). Do the authors mean that these sites are indeed in the transcript, or do they refer to the promoter area, upstream of the transcriptional start site?

In the same section, (Figure 5) CHIP experiments are done using C2C12 cells. Are these cells over-expressing HA-tagged β-catenin, as was done in Figure 5, or non-transfected cells?

A full analysis of the role of β-catenin-TCF on the Slit2 promoter would include mutating the binding sites and test for requirements. This is not unfeasible.

*Reviewer #3*:

Wu et al report an interesting study on a plausibly novel role of Slit2 in presynaptic differentiation in neuromuscular junction. The authors used a clever genetic trick to replace β-catenin in muscle cells with mutant versions that either cannot mediate cell adhesion or cannot activate downstream genes and showed that a transcription based mechanism in the muscle cells requiring β-catenin regulates the presynaptic differentiation in the motoneurons. From there, they screened for transcription targets of β-catenin in muscle cells. Slit2 was one of the many targets. Using a transgenic line expressing Slit2 in muscle cells in β-catenin mutant mice, the authors were able to rescue the phenotypes of overgrowth of the presynaptic terminals. Recombinant Slit2 can also induce synaptophysin puncta of spinal cord explants. Although the finding that Slit2 may be a novel retrograde signal for synapse formation is interesting, the current study does not provide sufficient evidence for this conclusion. There is also some concerns about the approach that is used, which involves overexpression of β-Catenin and Slit2.

1) Genetic evidence that Slit2 is important for NMJ development is lacking. Slit2 is one of the many downstream factors of β-catenin. Overexpressing Slit2 in β-catenin mutant mice can rescue the overgrowth of presynaptic terminals. However, the authors did not show or discuss loss of function results. It is not clear whether other factors are more important in vivo.

2) The phenotype of β-catenin knockout in muscle cells lead to overgrowth of motoneuron terminal arbors. Slit2 is an inhibitor of axon outgrowth. It is possible that overexpressing Slit2 reduces the arbor growth by enhancing growth inhibition. And the synaptic differentiation phenotypes may be due to a secondary effect to axon growth because the axons were not able to find the target area for efficient synapse formation. Therefore, Slit2 KO phenotype will be informative.

3) The promoters for expressing both β-catenin and Slit2 are much stronger than their endogenous promoters. Therefore, β-catenin and Slit2 are overexpressed several fold more (Figure 1—figure supplement 1, Figure 6—figure supplement 1). This could lead to nonphysiological effects.

---

## [Author Response]

Reviewer #2:

*[…] To start with a general question, if Slit2 is a transcription target of the β-catenin-TCF complex, it would be likely that β-catenin is activated by a Wnt signal, as the majority of cases of β-catenin acting as a transcription factor are due to Wnt signals. In the muscle, this would be interesting to know, in particular where the Wnts are expressed. In*
Figure 4*, some Wnts are tested for overall expression but this is a limited survey. As a locally acting signal, the Wnt protein could contribute, in a deterministic way, where on the muscle fiber β-catenin and subsequently Slit2 would be expressed, setting up the site of the NMJ. It could of course be that Wnt is coming from the incoming neuron. I realize that possible local signaling in a muscle fiber is complicated by the multinucleated nature of the muscle, but it still remains an important question whether β-catenin is locally activated. The latter point could be examined by a simple stain for β-catenin in the muscle while the use of Wnt/TCF reporter mice could indicated local transcriptional activation*.

We thank the reviewer for the question whether β-catenin is locally activated. As suggested, we performed two additional experiments to address this great question and results are presented in new, Figure 2—figure supplement 1. First, whole-mount staining of muscle fibers with β-catenin antibody indicates that β-catenin in synaptic nuclei appeared to be higher than that in extra-synaptic regions. Second, we characterized *Axin2*-nlacZ mice, where the nuclear-localized β-galactosidase (nlacZ) DNA was inserted in the *Axin2* gene, a target of Wnt/β-catenin signaling. Expression of β-galactosidase in *Axin2*-nlacZ mice is controlled by *Axin2* promoter has been used to as a reporter of Wnt or β-catenin activity (40). As shown in Figure2-supplemental Figure 2, β-galactosidase was detectable in synaptic nuclei, not in extrasynaptic nuclei. These results are in agreement with the notion of active Wnt/β-catenin signaling in synaptic nuclei at the NMJ and provide additional support to the model. These results are presented in Figure 2—figure supplement 1.

*The survey of secreted molecules (*Figure 4*) shows that apart from Slit2, several others seem to be regulated by the absence of β-catenin as well. It is not clear why these genes were excluded from further analysis*.

Sorry for being unclear. A better rationale to focus on Slit2 is now discussed in the revised Discussion (paragraph four).

*Regarding the experiments that show a role of β-catenin-TCF in the activation of Slit2, the text mentions binding sites in the 5'UTR (subsection “Identification of Slit2 as a retrograde target”). Do the authors mean that these sites are indeed in the transcript, or do they refer to the promoter area, upstream of the transcriptional start site*?

Sorry for being unclear. Sites were identified upstream from the transcription initiation site. This is now described in revised Results.

*In the same section, (*Figure 5*) CHIP experiments are done using C2C12 cells. Are these cells over-expressing HA-tagged β-catenin, as was done in*
Figure 5*, or non-transfected cells*?

We apologize for being unclear. ChIP experiments were done in C2C12 cells transfected with HA-tagged β-catenin. This information is now provided in the revised Methods.

*A full analysis of the role of β-catenin-TCF on the Slit2 promoter would include mutating the binding sites and test for requirements. This is not unfeasible*.

We agree with the reviewer that a full analysis of the Slit2 promoter could be informative. This paper focuses on functional evidence that Slit2 serves as a β-catenin-regulated retrograde factor for presynaptic differentiation. We demonstrate, first, that Slit2 levels were reduced in β-catenin mutant muscles. Second, β-catenin associates with TCF cis-elements in the promoter region of the Slit2 gene. Third, Slit2 expression in muscle cells is promoted by elevating β-catenin. Slit2 was concentrated at the NMJ. Fourth, transgenic mice expressing Slit2 specifically in the muscle was able to diminish presynaptic deficits by β-catenin mutation, including poor innervation, reduced vesicle density and active zone, and depressed mEPP. Finally, Slit2 immobilized on beads was able to induce the synaptophysin puncta of axons of spinal cord explants.

We hope that the reviewer agree that a full analysis of the Slit2 promoter would be beyond the scope of this paper.

Reviewer #3:

*1) Genetic evidence that Slit2 is important for NMJ development is lacking. Slit2 is one of the many downstream factors of β-catenin. Overexpressing Slit2 in β-catenin mutant mice can rescue the overgrowth of presynaptic terminals. However, the authors did not show or discuss loss of function results. It is not clear whether other factors are more important in vivo*.

These points are well taken. First, as suggested, phenotypes of Slit2 mutant mice were discussed in the third paragraph of Discussion. Second, we agree with the reviewer that other factors may be involved. In fact, Slit2 expression in muscles is unable to rescue all defects in β-catenin mutant mice. This point is now indicated in the revised Discussion (fourth paragraph).

*2) The phenotype of β-catenin knockout in muscle cells lead to overgrowth of motoneuron terminal arbors. Slit2 is an inhibitor of axon outgrowth. It is possible that overexpressing Slit2 reduces the arbor growth by enhancing growth inhibition. And the synaptic differentiation phenotypes may be due to a secondary effect to axon growth because the axons were not able to find the target area for efficient synapse formation. Therefore, Slit2 KO phenotype will be informative*.

Our paper provides solid evidence that muscle Slit2 regulates presynaptic differentiation.

Having demonstrated that the transactivation activity of β-catenin is critical, we reasoned that muscle β-catenin is likely to regulate presynaptic differentiation by controlling the expression of muscle-derived proteins. This notion was supported by results of additional experiments for the revision—that β-catenin and Wnt signaling reporter were higher in synaptic nuclei at the NMJ than those in extra synaptic areas. We reasoned that such a protein is secretable or has a transmembrane domain and its expression is down regulated in β-catenin mutant muscle. A screen for proteins that fulfill these criteria led to the identification of Slit2.

Next, in a set of in vivo and in vitro experiments we demonstrated, first, that Slit2 levels were reduced in β-catenin mutant muscles. Second, β-catenin associates with TCF cis-elements in the promoter region of the Slit2 gene. Third, Slit2 expression in muscle cells is promoted by elevating β-catenin. Slit2 was concentrated at the NMJ. Fourth, transgenic mice expressing Slit2 specifically in the muscle was able to diminish presynaptic deficits by β-catenin mutation, including poor innervation, reduced vesicle density and active zone, and depressed mEPP. Finally, Slit2 immobilized on beads was able to induce the synaptophysin puncta of axons of spinal cord explants.

Together, these observations make a reasonably compelling case for Slit2 as a β-catenin downstream factor for presynaptic differentiation.

We agree with the reviewer that a full study of Slit2 mutant mice could be more informative. However, we hope that the reviewer would agree that such a study is beyond the scope of this paper, considering that the study has utilized multiple transgenic strains and muscle-specific mutant lines and the paper has 8 figures, each with multiple panels, plus several supplemental figures.

*3) The promoters for expressing both β-catenin and Slit2 are much stronger than their endogenous promoters. Therefore, β-catenin and Slit2 are overexpressed several fold more (*Figure 1—figure supplement 1*,*
Figure 6—figure supplement 1*). This could lead to nonphysiological effects*.

Agree. However, to achieve muscle-specific expression prior to NMJ formation, HSA remains to be an ideal promoter. The other well-characterized muscle-specific promoter, MCK, is not active until neonatal stage and is thus unsuitable for studying NMJ formation. We would like to point out that our conclusion is not simply based on transgenic expression, rather a comprehensive set of in vivo and in vitro experiments. A parsimonious explanation of the results is that Slit2 serves a retrograde factor downstream of muscle β-catenin for presynaptic differentiation.